# Dietary Habits and Tusk Usage of Shovel-Tusked Gomphotheres from Florida: Evidence from Stereoscopic Wear of Molars and Upper and Lower Tusks

**DOI:** 10.3390/biology11121748

**Published:** 2022-11-30

**Authors:** Gina M. Semprebon, Jeanette Pirlo, Julia Dudek

**Affiliations:** 1Department of Biology, Bay Path University, Longmeadow MA 01106, USA; 2Department of Biological Sciences, College of Science, Stanislaus Campus, California State University, Stanislaus, CA 95382, USA

**Keywords:** microwear, paleodiet, shovel-tusker, tusk, proboscidean

## Abstract

**Simple Summary:**

The shovel-tusked gomphotheres have long intrigued paleontologists and the general public as they are one of the most notable of all elephant ancestors. This is due to their highly distinctive and broad, flattened lower tusks that look remarkably like the head of a shovel. There has been much speculation regarding how these bizarre mammals lived as well as the possible functions of their tusks. This study aims to provide evidence as to their ancient diet as well as the likely functions of their upper and lower tusks by examining scars left in their molar as well as the scratch patterns left on their tusks. We show that the Florida shovel-tusked gomphotheres browsed on leaves and twigs, and thus, most likely occupied a dietary niche similar to living forest elephants. We demonstrate that they used their tusks for a wide variety of purposes. Upper tusks were used for scraping and slicing, whereas lower tusks were used for shoveling in some forms and scraping in others. The obvious importance of tusks in the life of proboscideans such as the shovel-tuskers underscores the vital importance of saving modern elephant populations from poachers that callously steal their tusks for profit.

**Abstract:**

The paleodiet of the shovel-tusked gomphotheres from Florida (*Amebelodon floridanus*, *Konobelodon britti*, and *Serbelodon barbourensis*) was assessed via microwear analysis of molar dental enamel and compared to a large database of both extant proboscideans and ungulates. Scratch and pit results show a consistent browsing signal in *A*. *floridanus*, *K*. *britti* and *S*. *barbourensis*. Fossil results are more similar to those of the extant *Loxodonta cyclotis* than to *Loxodonta africana* or *Elephas maximus*, the latter two taxa exhibiting a mixed feeding result. Scratch width scores are high in all three shovel tuskers as well as in the extant proboscideans indicating the ingestion of some coarse vegetation, most likely bark, and twigs. Gouging is relatively low in *A*. *floridanus* and *S*. *barbourensis*. Only *K*. *britti* has levels of gouging approximating that seen in extant elephants. Large pitting is relatively low in both fossil and extant forms although *L*. *cyclotis* has higher levels of large pitting including more puncture-like pits seen with fruit and/or seed consumption. A variety of scratch patterns indicating variation in tusk usage behavior was found. Some *Serbelodon* and *Konobelodon* mandibular tusks exhibited digging behavior, although *Konobelodon* digging behavior was much more common and obvious, whereas *Amebelodon* mandibular tusks did not exhibit digging behavior and were more likely used for stripping and scraping. Unusual distal tusk wear was found in *Amebelodon* and *Serbelodon* most likely due to stripping off tree bark. Upper tusk usage varied with all three fossil species exhibiting scraping and/or cutting behavior. Results indicate that shovel-tusked gomphotheres from Florida occupied a narrow dietary niche but employed a variety of strategies to obtain the vegetation that they consumed.

## 1. Introduction

### 1.1. Background

Gomphotheres were a group of extinct, large elephant-like proboscideans that had an almost global distribution during the Miocene, Pliocene, and Pleistocene except for Antarctica and Australia. By the early Miocene, they made it into Eurasia from Africa [1]. By the middle Miocene, they reached North America [2], after which they arrived in South America because of the Great American Interchange [3]. Gomphothere molars are either trilophodont or tetralophodont, bunodont and generally low-crowned. Therefore, they are regarded as most likely having been opportunistic browsers [4,5,6]. Gomphotheres are generally thought to be a paraphyletic assemblage (e.g., [7]) and their phylogeny is still unclear, but they were a very diverse and successful group. Four tusks, two upper and two lower are the ancestral condition for the group. The upper tusks were curved and oriented in an outward and downward direction [8], while their lower tusks were more spatulate in appearance and procumbent [8]. Their retracted facial and nasal bones led early paleontologists to reconstruct them as having elephant-like trunks ([9] “Figure 1A”) Gomphotheres peaked in terms of their diversity in the late Miocene but went extinct in the Pleistocene.

The shovel-tusked gomphotheres are some of the most distinctive of all proboscideans mostly due to their dorso-ventrally flattened and broad mandibular tusks (Figure 1B). Osborn [12] fueled the intrigue for shovel-tuskers by both the general public and paleontologists by likening the Asian specimens to the head of a shovel. The group has been given family status as the family Amebelodontidae (e.g., [13,14]) or thought of as the subfamily Amebelodontinae (e.g., [2,7,15,16]). Amebelodonts are known mostly from the Miocene Epoch.

The shovel-tusked gomphotheres registered from Florida will be studied.

### 1.2. Fossil Localities

Figure 2 shows the Florida localities where the shovel-tusker taxa analyzed in this study were found. *A*. *floridanus* is from the Mixson’s Bone Bed (early Hemphilian) and Tyner Farm (early Hemphillian). Mixson’s Bone Bed is a rather massive clay bed which typically indicates a quiet water environment (no marine vertebrates have been reported however [17,18]. Tyner Farm is an ancient sinkhole which has been characterized as a mostly terrestrial closed environment [19]. *Konobelodon britti* is from the Moss Acres Racetrack (late early Hemphillian), also a massive clay but fine-grained deposited within a depression within the limestone bedrock. This depositional environment is thought to represent an ancient lake or pond with very still water. *Serbelodon barbourensis* is from the Love Bone Bed (latest Clarendonian). Webb et al. [20] described the Love Site as a graded fluvial deposit within a bioclastic limestone unit most likely representing a single depositional cycle of an ancient stream or small river. We recognize that there is a possibility that some or all of the gomphotheres studied here could have inhabited a particular locality at the same time.

### 1.3. Aims of the Study

The purpose of this study is twofold: (1) to explore the paleodietary niche ecology of Florida shovel-tusked gomphotheres; (2) to compare microwear patterns of *A*. *floridanus*, *S*. *barbourensis*, and *K*. *britti* to each other and to those of extant elephants.; (3) to determine the likely functions of the upper and lower tusks in Florida shovel-tuskers. The hypotheses to be tested include: (1) *A*. *floridanus*, *S*. *barbourensis*, and *K*. *britti* share a common dietary strategy—browsing on leaves and twigs; (2) *A*. *floridanus*, *S*. *barbourensis*, and *K*. *britti* occupy a dietary niche more similar to more closed habitat African elephants and to Florida mastodons than to open habitat African and Asian elephants; (3) Florida shovel-tuskers used their tusks in a variety of ways.

### 1.4. Microwear

Microwear has been utilized for more than three decades to assess food scars left in dental enamel by plant phytoliths or by exogenous dust or grit coating the surface of vegetation (e.g., [22,23,24,25,26,27,28,29,30,31]) and to reconstruct the diet of many different mammals including mammoths, mastodons, and other proboscideans [32,33,34]. Microwear has proven to be valuable because it turns over relatively rapidly. The value of this rapid turnover should not be underestimated as it allows for insight into the dietary behavior of the last days, or weeks before an animal’s death—the so-called “Last Supper Effect” [35]. Because of this, it allows for a window into the short-term dietary behavior of a taxon rather than the deep-time adaptation of gross tooth form. Rivals and Semprebon [36] have stressed that this allows for insight into what an animal was actually eating at the time of its death despite what it might have been adapted to eat over deep time. Thus, microwear is a direct source of what an animal’s dietary behavior has been at a particular time rather than cumulative wear over its lifetime (e.g., mesowear) or through its deep-time ancestry (e.g., hypsodonty). While the latter two techniques are highly valuable, microwear reveals daily, seasonal, or regional variations in diet that gross tooth methods generally cannot reveal [36,37,38]. The technique employed in this study to allow for the comparison of results on fossil shovel-tuskers to a very large comparative microwear database comprised of extant ungulates and proboscideans of known diets compiled by a single observer (GMS) to minimize error [24,34].

## 2. Materials and Methods

### 2.1. Sample

Molar teeth of *Amebelodon floridanus*, *Konobelodon britti* and *Serbelodon barbourensis* were sampled from the collections of the Florida Museum of Natural History, Florida, USA. All specimens were screened for potential taphonomic alteration of enamel surfaces using a stereomicroscope. Those displaying taphonomic defects were removed from the analysis after King et al. [39] as well as heavily worn teeth. A total of 58 molars were found to be suitable for analysis and analyzed using the low magnification microwear technique by a single observer (GMS) who also analyzed the extant ungulate and extant proboscidean comparative samples. In addition, 14 tusks were found to be suitable for analysis and examined under 35 times magnification (Table 1 and Table 2).

In North America, the genus *Amebelodon* is known from the type species *Amebelodon fricki* from the Great Plains and *Amebelodon floridanus* from the Great Plains and Florida [1]. Other North American species that were formerly placed in this genus have now been considered synonyms of one of these or referred to different genera. For example, *Konobelodon* was originally thought to be a subgenus of *Amebelodon* [40] but was given full generic rank by Konidaris et al., [41]. In this study, we will refer to the former *Amebelodon britti* as *Konobelodon britti*. The species is known from Texas, Kansas, and Florida and is considered to possibly be one of the largest land mammals to have ever lived in North America [40,42]. The type locality is the Moss Acres Racetrack Site (Marion County, Florida [41]). In this study, we will refer to the former *Amebelodon barbourensis* [8] as *Serbelodon barbourensis* [43,44]—the *A*. cf. *barbourensis* of Webb et al. [20]. The *S*. *barbourensis* type locality is Xmas Quarry from the Ash Hollow Formation of Nebraska. It is known from Nebraska and Florida.

### 2.2. Microwear Technique

The methodology of [24,25] was used to clean, mold, cast, illuminate and examine molar surfaces at 35 times magnification with a Zeiss Stemi-2000C stereomicroscope. The analysis was made from the central portion of the enamel of the second transverse ridge on upper or lower teeth (grinding facet “Figure 1C”). External oblique illumination was used to visualize microscopic enamel scars using a high intensity fiber-optic light source (M1-150 Dolan-Jenner light) directed across the casts’ surface at a shallow angle to the occlusal surface. An ocular reticle (0.4 mm square area or 0.16 mm^2^) was used to count the average number of pits (rounded features), average number of scratches (elongated features) and average number of large pits (regular edged pits with low refractivity or dark and at least twice the diameter of highly refractive or shiny, small white pits per taxon). In addition, it was recorded if gouges (large depressions with irregular edges) were present or absent per microscope field within the 0.4 mm square area (after [24]).

Results on fossil teeth were compared with those from extant ungulates and proboscideans [24] to determine the dietary categories of browser versus grazer. Scratch textures were classified as being either fine, coarse, a mixture of fine and coarse, a mixture of coarse and hypercoarse, or hypercoarse scratch types per tooth surface using differential light refractivity (after [25]). A scratch width score (SWS) was obtained by assigning a score of 0 to teeth with mostly fine scratches per tooth surface, 1 to those with a mixture of fine and coarse types of textures, 2 to those with predominantly coarse scratches, 3 to those with a mixture of coarse and hypercoarse textures, and 4 to those with predominantly hypercoarse scratches. To help distinguish mixed feeders that alternate between browse and grass, the percentage of individuals within a taxon that possess raw scratch values falling into the low scratch range (i.e., between 0 and 17) were calculated [37]. Univariate statistics, ANOVAs and Tukey’s post hoc test for honest significant differences were calculated using PAST 4.11 software [45].

## 3. Results

Table 3 shows the summary statistics for molar microwear for the Florida shovel tuskers analyzed here. The dietary assignment is the same (i.e., browser) for all three taxa. Raw data are shown in Appendix A.

An ANOVA (Appendix A) indicates: (1) that *A*. *floridanus* pit numbers are significantly different (higher) from those of *Serbelodon barbourensis* (Tukey’s HSD Test, *p* = 4.188 × 10^−5^), significantly different (higher) from *Konobelodon britti* (Tukey’s HSD Test, *p* = 7.69 × 10^−5^), significantly different (higher) from *Loxodonta africana* (Tukey’s HSD Test, *p* = 0), and significantly different (higher) from *Elephas maximus* (Tukey’s HSD Test, *p* = 4.67 × 10^−12^); (2) *A*. *floridanus* scratch numbers are significantly different (lower) from those of *Loxodonta africana* (Tukey’s HSD Test, *p* = 3.22 × 10^−9^) and significantly different (lower) than those of *Elephas maximus* (Tukey’s HSD TEST, *p* = 1.891 × 10^−5^). (3) *A*. *floridanus* large pit numbers are significantly different (lower) from those of *Serbelodon barbourensis* (Tukey’s HSD TEST, *p* = 0.01467). *Serbelodon barbourensis* scratch numbers are significantly different (lower) from those of *Loxodonta africana* (Tukey’s HSD TEST, *p* = 0.01467).

Figure 3 shows raw scratch and pit results for extant proboscidean molars plotted in reference to extant ungulates (ellipses) and Proboscidea of known diets. It is clear from Figure 3 that extant elephants have a diverse diet (i.e., are mixed feeders on browse and grass) although the forest elephant (*Loxodonta cyclotis*) has results indicating less grass in its diet than the other taxa.

Figure 4 shows fossil shovel-tusker raw scratch plots. It is evident in Figure 4 that the shovel-tusked gomphotheres analyzed here have a less diverse diet than extant elephants (Figure 3). That is, they are primarily browse-dominant low scratch feeders. The distributions are unimodal and consistently in the low scratch range as is seen in extant leaf—dominated browsers (see [24]).

Figure 5 shows a bivariate scratch/pit plot of the average scratch versus pit results of the fossil shovel-tusker taxa compared to extant ungulates (convex hulls) and living proboscidean average scratch/pit values. All three shovel-tuskers fall in the extant leaf browsing ecospace and close to the extant *Loxodonta cyclotis*, indicating a less abrasive and more restrictive dietary regime than those of *Loxodonta africana* and *Elephas maximus*.

Figure 6 depicts a box plot of scratch width scores for fossil shovel-tuskers compared to those for extant ungulate browsers, grazers and mixed feeders and proboscideans and a photomicrograph taken at 35× with a stereomicroscope of a typical enamel surface of the fossil shovel-tuskers analyzed in this study (data from [34]). Figure 6A demonstrates that typical extant leaf-dominated ungulate browsers as a group possess mostly finely textured scratches; whereas typical grazers have fewer fine and more coarse scratches and show no overlap with leaf browsers. Ungulate mixed feeders overlap both leaf browsers and grazers because they consume browse and grass regionally or seasonally. It is clear that extant proboscideans are unique in that they have very large mean scratch width scores when compared to other herbivorous mammals. Figure 6B shows these very wide and deep (i.e., hypercoarse) scratches found typically on proboscidean molar enamel and rarely on other herbivores with the exception of some hard object processors (such as hard fruit/seed consumer [25]).

Figure 7A shows the longitudinal lateral enamel band typically found on shovel-tusker and most other gomphothere upper tusks. Figure 7B shows scratches oriented perpendicular or angled relative to the enamel band on the ventral surface of the upper right tusk of *K*. *britti* (UF 69998); Figure 7C shows scratches oriented parallel to the enamel band of the ventral surface of the upper right tusk of *A*. *floridanus* (UF 212305); Figure 7D depicts a strange wear facet on the tip of the upper right tusk of *A*. *floridanus* (UF 212305) with vertical deep scratches along the ventral rim of the facet indicating likely scraping behavior.

Table 4 gives the results from the examination of the upper tusks of *A*. *floridanus*, *K*. *britti* and *S*. *barbourensis*. All the tusks of *A*. *floridanus* examined had scratches that were mostly parallel to the upper tusk enamel band. Most of the tusks examined for *K*. *britti*, on the other hand, had scratches mostly perpendicular to the enamel band. In addition, two upper tusks of *A*. *floridanus* exhibited tip wear while tip wear was not observed on the upper tusks of *K*. *britti*.

Figure 8 and Table 5 offer results on the mandibular tusk wear of *A*. *floridanus*, *K. britti* and *S*. *barbourensis*. Unusual mandibular tusk tip wear and tip facets in *A*. *floridanus* (Figure 8A) and *S*. *barbourensis* (Figure 8B) were observed. Figure 8C shows a non-polished ventral surface of a lower tusk of *A*. *floridanus* while Figure 8D–F show polished wear (ventral) in an *S*. *barbourensis* lower tusk (D), polished dorsal wear in a *K*. *britti* lower tusk (E), and ventral lower tusk wear in *K*. *britti* (F). *A*. *floridanus* showed no dorsal and ventral wear facets or polishing but occasional unusual tip wear and facet (Table 5). *K*. *britti* exhibited no dorsal and ventral wear facets or polishing or tip wear, but *S*. *barbourensis* showed dorsal and ventral wear facets and tip wear and facets but no dorsal and ventral polishing (Table 5).

## 4. Discussion

### 4.1. Value of Microwear

An important thing to consider in any paleodietary reconstruction is that taxa from different localities and/or time periods would be confronted by a variety of habitats that could influence their dietary behavior regardless of their molar morphology acquired over deep time. Microwear is valuable precisely because it is direct evidence of what food items were actually consumed by an animal at the time of their death despite what that animal might have been optimally adapted to be eating because of its deep past. This can play out in two ways. Firstly, a highly hypsodont animal living today such as the pronghorn antelope (e.g., *Antilocapra americana*) may have evolved from predecessors that engaged in a highly abrasive diet such as grass and/or lived in an open habitat where grit encroached on food items—increasing the abrasiveness of its food regime [37,46]. This animal may today eat mostly browse, but its tooth crown height was most certainly influenced by this highly abrasive dietary regime in its evolutionary past. Microwear analysis, therefore, is pivotal to detect this browsing behavior because crown height alone would lead to an entirely different dietary classification. The converse situation could also be true and is more pertinent to this study. That is, brachydont and bunodont animals such as gomphotheres, have been shown to engage in a variety of trophic behaviors despite being optimally adapted to feed on soft vegetation or browse ([5,6,46].

### 4.2. Fossil Site Paleoecology

*Serbelodon barbourensis* was recovered from the Love Bone Bed (LBB) fauna from Alachua County. The LBB has been dated (biostratigraphically) to the latest Clarendonian and represents a high energy stream channel with fossils drawn from adjacent source areas such as wet prairies and swamps that are still abundant in Florida today [20]. Vertebrate fossils from the Love site suggest three terrestrial habitats: forest, streambank, and open country. Important to the results of this study, the LBB is known for its unusually abundant forest-dwelling species compared to many late Miocene High Plains fluviatile sites [20]. This is particularly true of the browsing ungulates from the LBB. Graham [47] considers that the Gulf Coast Savanna region had more extensive mesic forests in contrast to the Great Plains most likely providing a favorable environment for browsing mammals.

*Amebelodon floridanus* was recovered from Mixson’s Bone Bed (Alachua Formation) and Tyner Farm. In 1884, the United States Geologic Survey received fossils from a plantation owner in Levy County Florida, J.M. Mixson. These, and others excavated through 1890, were sent to Leidy for examination. Most of the bones were made up of teeth and isolated bones. Large fossils were fragmentary and very few articulated skeletons were found amid the clay bed [48]. Of note, fossils of what the author was calling “Mastodon” were found, but these “Mastodon” fossils were very different from the usual fossils around the United States [48]. These strange mastodons turned out to be *Amebelodon floridanus.* They also found fossils from *Teleoceras proterum, Aphelops malacorhinus, Cormohipparion plicatile, Thinobadistes segnis, Aepycamelus major,* and *Tapirus webbi.* A special note was made about the absence of carnivore remains, or bones of small mammals [48], although carnivore fossils were found in the second round of excavations. The second wave of excavations at Mixson’s occurred between the 1930s–1940s, with “thousands” of fossils being collected and catalogued at the American Museum of Natural History [49]. Mixson’s Bone Bed, as the site came to be called, was biochronologically dated to the Late Miocene (Hh1, ~8–9 Ma) and is important to the understanding of environment and climate of the southeastern United States because it was the first Neogene terrestrial vertebrate site in the United States [49]. Further examination of the site determined it was a quiet water environment, similar to a lake, as fossils of turtles and alligators are common. Tyner Farm (early Hemphillian (Hh1, ~7.5 myo) is an ancient sinkhole in North Central Florida. It was discovered by landowners, Bruce and Allan Tyner as they were plowing the property to plant peanuts. The site was excavated until the limestone floor of the sinkhole was reached. At least 40 different species were collected, including microfossils of small mammals, reptiles, and amphibians, as well as large mammals *like Borophagus orc, Tapirus webbi, Teleoceras hicksi,* and *Amebelodon floridanus* [19]. Although some aquatic species were recovered, they are rare, suggesting that the sinkhole preserves a mainly terrestrial environment. Congruent with our browsing results for the Florida shovel-tuskers, it has been determined based on the herbivorous taxa from the site, it has been determined that Tyner Farm was relatively closed in with brush and trees, and very little open grassland [19]. This paleoenvironment is supported by arboreal taxa like tree squirrels, and other large browsing herbivores like *Thinobadasties wetzeli, Nannipus aztecus, Cormohipparion plicatile, Hipparion tehonense, Aphelops maalcorhinus,* and *Aepycamelus major* [50].

*Konobelodon britti* was recovered from the Moss Acres Racetrack (Marion County, Florida). Faunal lists published by [51,52] indicate a late early Hemphillian age for Moss Acres Racetrack. The fauna is part of the Alachua Formation and was derived from a large clay deposit (Great Interchange). Lambert [42] considers the fauna as an unmixed local community with no evidence that specimens were transported by water from other localities to be mixed in with the remains of animals that died at the site. *Konobelodon britti* from Moss Acres (which is its type locality) represents the earliest record of *Konobelodon* which eventually became extinct in the late Hemphillian [40]. Moss Acres has yielded an impressive fauna of large herbivores dominated by mammals such as *Konobelodon* and represents one of the richest localities in the world for this taxon [42]. Lambert [42] reports both abundant grass and trees at Moss Acres Racetrack and believes that North Florida possessed a savanna landscape during the late early Hemphillian. Webb [53] assigned *Konobelodon* to a browser guild using estimated body weight versus molar volume. Of all of the browsers present in the fauna, Lambert [40] considered *Konobelodon britti* as probably the most generalized in a dietary sense, thus eating whatever browse was available including tree bark. Lambert [42] characterizes the Moss Acres Racetrack flora as resembling that of a modern woodland in the southeastern United States with exotic tropical taxa absent.

It is clear from the reconstructions of the localities in Florida occupied by the shovel-tusked gomphotheres in this study, that trees were definitely available in these landscapes and browsers were abundant in these localities. This scenario is entirely consistent with the molar microwear results from this study. Raw scratch plots (Figure 4) of all three fossil species place them as browse-dominant feeders (i.e., unimodal, low scratch range results). Likewise, average scratch/pit results (Figure 5) place all three shovel-tuskers in the extant leaf browsing ecospace. Finally, uniquely large scratch width scores for fossil shovel-tuskers place them with extant proboscideans that consume bark (Figure 6). Solounias and Semprebon [24] have shown that these hypercoarse (very wide and deep) scratches are typical in extant bark consumers.

### 4.3. Molar Microwear

One of the aims of this study was to explore the paleodietary niche ecology of Florida shovel-tusked gomphothere by comparing molar microwear patterns of *A*. *floridanus*, *S*. *barbourensis*, and *K*. *britti* to each other and to those of extant elephants. We tested the hypotheses that *A*. *floridanus*, *S*. *barbourensis*, and *K*. *britti* share a common dietary strategy—browsing on leaves and twigs and that *A*. *floridanus*, *S*. *barbourensis*, and *K*. *britti* occupy a dietary niche more similar to more closed habitat African forest elephants than to open habitat African savannah and Asian elephants. Microwear results here also are congruent with gross molar morphology which is bunodont and brachydont (Figure 1C). However, Rivals and Semprebon [36] caution that an animal’s gross molar morphology has been shaped over deep time to eat a particular diet, but that does not mean that it will not vary its diet depending on local favored food availability at a given time and/or place. Microwear analysis is valuable precisely because it allows for insight into what an animal was really eating at a given time or place at the time of its death despite what it might have been adapted to eat over most of its evolutionary history. Because of their brachydont and bunodont molar morphology, gomphotheres have often been assumed to be browsers. However, isotopic analyses of the South American gomphothere *Notiomastodon platensis* found a wide dietary range for this taxon except for two localities where the taxon was engaging in an exclusive C4 diet [5]. In this study, it was apparent that the animals were engaged in diets that corresponded to the latitudinal gradient of C3/C4 grasses [5]. Stereomicrowear results placed *N*. *platensis* within the extant mixed feeder morphospace.

Our results indicate that the shovel-tusked gomphotheres from Florida (Table 3) consistently show a mostly browsing dietary signature regardless of the fossil taxon or locality that they were recovered from (see Figure 5 and 0–17% low scratch results in Table 3). Evenso, *A*. *floridanus* total pit numbers are significantly higher from those of *Serbelodon barbourensis* and *Konobelodon britti*. Browsers often exhibit a higher range of pit numbers from those taxa that are mixed feeders on grass and browse [24] so it is not surprising that *A*. *floridanus* would have higher pit numbers than the extant *L*. *africana* and *E*. *maximus*. The fact that *A*. *floridanus* has significantly higher total pit numbers than *S*. *barbourensis* and *K*. *britti*, however, is interesting and may indicate the consumption of a coarser type of browse in the former. *A*. *floridanus* large pit numbers are significantly lower from those of *S*. *barbourensis*, however, and *S*. *barbourensis* is notable for having more large pits and gouges than *A*. *floridanus* a finding that may indicate the occupation of a slightly more open habitat in the latter versus the former. *A*. *floridanus* and *S*. *barbourensis* have significantly less scratches than extant elephants which was expected due to their mostly browsing results. Results for *K*. *britti* should be taken with caution as the specimen number was very low for this taxon (Appendix A).

The taxa analyzed here appear to have been dedicated browsers, corroborating our first hypothesis that they shared a common browsing dietary strategy. These results are consistent with a study that examined the C_3_ vs. C_4_ ratios in tusks to reconstruct the dietary habits of North American gomphotheres which found that they stayed in wooded habitats and ate from trees and shrubs [4]. Zhang et al. [54] reported similar results in China. Our results also are in line with our second hypothesis which was to test whether *A*. *floridanus*, *S*. *barbourensis*, and *K*. *britti* occupy a dietary niche more similar to more closed habitat African forest elephants than to open habitat African savannah and Asian elephants. Figure 5 demonstrates that *A*. *floridanus*, *S*. *barbourensis*, and *K*. *britti* have average scratch and pit results that cluster closer to *L. cyclotis* in the extant browser ecospace than to *Loxodonta africana* or *Elephas maximus* corroborating our hypothesis that the fossil forms most likely occupied a dietary niche more similar to more closed habitat extant elephants like *L*. *cyclotis* than to more open forms like *L*. *africana* and *E*. *maximus*.

### 4.4. Tusk Microwear

Wear patterns on upper and lower tusks have been used to elucidate the feeding behaviors of gomphotheres, including the shovel-tusked gomphotheres from this study. Lambert [42,55] examined Florida shovel- tusker tusk wear and reported strong evidence that these taxa were likely feeding largely on trees, and in particular, their bark. This conclusion is totally supported by the examination of both molars and tusks in this study. Wear patterns on upper tusks have also been used to assess the particular method of attaining food in gomphotheres [55]. Lambert examined the functional role of upper tusks in feeding of typical gomphotheres [55]. As such, he discussed the fact that the upper tusks of such gomphotheres possess a strong enamel band on their lateral surfaces but that this enamel band is not found in extant elephants (Figure 7A). Lambert [55] postulates that this lateral enamel band forms a latero-posterior border to upper tusk wear facets and the enamel band is exposed by it such that a food substance moving from anterior to posterior along a facet would encounter the enamel band as a hard edge capable of cutting. Scratches parallel to the enamel band would therefore indicate slicing, whereas scratches perpendicular to the enamel band would indicate scraping. Lambert points out [55] that scraping must have occurred first in order to expose and probably maintain the enamel band edge and, thus, parallel scratches should be found accompanied by oblique scratches. Our results confirm this hypothesis. For upper tusks (Table 4 and Figure 7), both types of scratches are seen on each tusk examined although a predominant tusk scratch pattern was also observed. Thus, all three fossil taxa used their tusks for scraping and slicing. Lambert rightly cautions, however, that it is unclear whether or not scraping was a component of actual feeding behavior or merely necessary to produce a hard, raised edge along the enamel band.

Our results on lower tusks (Table 5 and Figure 8) indicate that only *S*. *barbourensis* mandibular tusks showed distinct dorsal and ventral wear facets consistent with encountering an abrasive substrate due to mandibular shoveling behavior. *K*. *britti* lower tusks exhibited polishing of dorsal and ventral tusk surfaces rather than wear facets and must have shoveled less abrasive materials. Finally, *A*. *floridanus* did not exhibit wear facets on its dorsal or ventral surfaces or polishing but did exhibit tip wear as did *S*. *barbourensis*. Lambert [55] posited that such tip wear that does not extend onto the dorsal or ventral surfaces is due to the scraping of mandibular tusks along the trunks of trees to strip off bark. The tightly packed parallel scratches observed on the tips of *A*. *floridanus* and *S*. *barbourensis* in this study are congruent with such activity especially because the scratches observed are hypercoarse in nature (see [24]). Like Lambert [55], the lack of clear wear dorsal and ventral facets and/or polished dorsal and ventral surfaces in *A*. *floridanus* suggest no shoveling behavior in this taxon but rather reliance on an unusual type of tip scraping wear. It is clear that shovel-tusked gomphotheres used their upper tusks in a variety of ways—for scraping, slicing, or even digging thus corroborating our third hypothesis.

Perez-Crespo et al. [56] inferred diets for many gomphotheres from those that inhabited the Americas during the Cenozoic (i.e., *Amebelodon, Cuvieronius, Gomphotherium, Rhynchotherium, Serbelodon,* and *Stegomastodon*) based on stable carbon isotope published data from tusk dental enamel and dentine. During the Barstovian and Clarendonian, these genera fed exclusively on plants, but the same genera in the Hemphillian consumed a mixed C_3_/C_4_ diet. *Rhynchotherium* in the Hemphillian apparently consumed a mixed feeding diet but switched to feed on C_3_ plants in the Blancan while *Stegomastodon* from this time period consumed a mixed C_3_/C_4_ diet and Lujanian specimens consumed C_3_ plants. *Cuvieronius* in the Irvingtonian, Rancholabrean and Lujanian consumed a mixed C_3_/C_4_ diet. Through their synthesis of available data from tusk enamel and dentine stable carbon isotopes Perez-Crespo et al. [56] suggest that American gomphotheres were primarily generalist feeders allowing them to occupy different areas of the continents and varying environmental conditions such as the shift from C_3_ ecosystems in the Barstovian and Clarendonian to mostly C_4_ ecosystems during the Hemphillian—possibly even also allowing them to survive glaciation episodes in the Pleistocene in some areas. However, while the Northern Hemisphere was becoming cooler in many areas as the Miocene progressed, it is probable that Florida was protected due to the warm waters of the Gulf of Mexico until the late Miocene when even Florida’s climate shifted to a drier and more seasonal one. This may be why the shovel tuskers analyzed here were all engaging in browsing on leaves, bark, and twigs. Of the taxa studied here, only *Amebelodon* survived into the late Hemphillian although it was a relatively rare taxon and its decline might have been caused by the widespread loss of woodland savanna from North America at this time [2].

## 5. Conclusions

The Florida shovel-tuskers have molar microwear scratch and pit results consistent with extant leaf browsers and low percentages of gouging and large pitting in their enamel compared to extant elephants indicating the occupation of more closed habitats then the extant forms. Scratch width scores were high in all of the fossil forms consistent with bark consumption which is common in extant proboscideans. All three fossil taxa used their upper tusks for scraping and slicing. Lower tusks of *S*. *barbourensis* and *K*. *britti* apparently were used for shoveling substrate while *A*. *floridanus* has no evidence of shoveling wear but rather scraping wear on distal tips. Florida shovel-tuskers apparently used their tusks for a variety of purposes.

## Figures and Tables

**Figure 1 biology-11-01748-f001:**
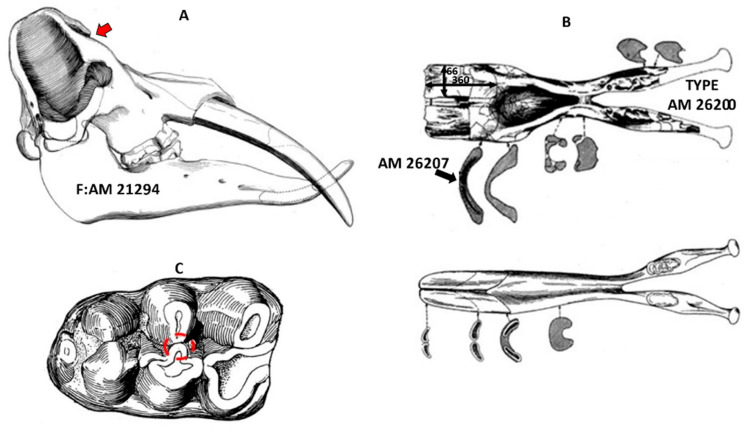
Gomphothere skull and tooth morphology. (**A**). *Gomphotherium productum* (modified from [8]; Taxon name as in [10]); (**B**). The lower jaws of *Amebelodon* (bottom) and *Platybelodon* (top) compared—note that these two shovel tuskers have relatively flat lower tusks rather than the typical oval or round tusks in other gomphotheres (from [11]); (**C**). M3 of *Serbelodon barbourensis* (F:A.M 21294 Modified from [8]). Red circle indicates area sampled on molars (grinding facet).

**Figure 2 biology-11-01748-f002:**
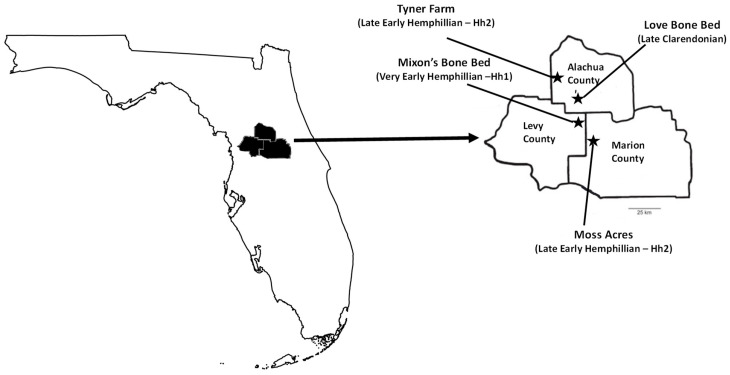
Fossil Localities. Clarendonian and Hemphillian fossil localities in Florida where shovel-tuskers were found (Modified Figure 1 from [21]; Map modified from GIS Geography.com/florida-county-map/, accessed on 25 May 2022).

**Figure 3 biology-11-01748-f003:**
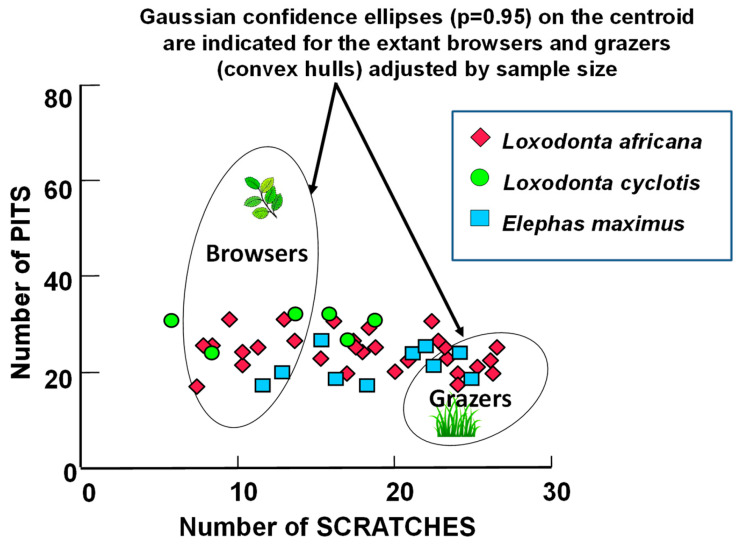
Raw scratch/pit results for extant elephant molars. Extant proboscidean raw scratch and pit results compared to extant ungulate scratch/pit microwear ecospaces. Extant browser and grazer ecospace data from [24]; extant proboscidean data from [34].

**Figure 4 biology-11-01748-f004:**
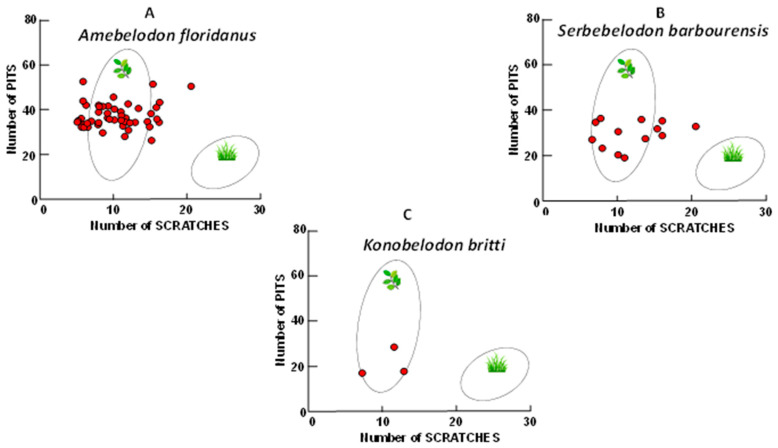
Fossil shovel-tusker raw scratch bivariate plots. Individual raw scratch and pit results of fossil shovel-tuskers compared to extant ungulate scratch/pit microwear ecospaces. (**A**). *Amebelodon floridanus* raw scratch/pit results. (**B**). *Serbelodon barbourensis* raw scratch/pit results; (**C**). *Konobelodon britti* raw scratch/pit results. Gaussian confidence ellipses (*p* = 0.95) on the centroid are indicated for the extant browsers on the left and grazers on the right (convex hulls) adjusted by sample size.

**Figure 5 biology-11-01748-f005:**
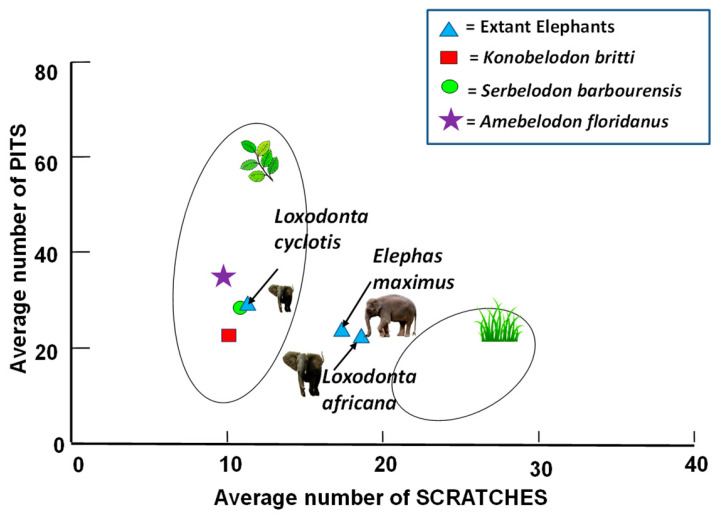
Fossil shovel-tusker average scratch/pit bivariate plots. Bivariate plot of the average number of pits versus the average number of scratches for the fossil and extant proboscideans analyzed graphed in relation to Gaussian confidence ellipses (*p* = 0.95) on the centroid for extant ungulate browsers (left) and grazers (right) (convex hulls) adjusted by sample size (ungulate data from [24]; elephant data from [34]).

**Figure 6 biology-11-01748-f006:**
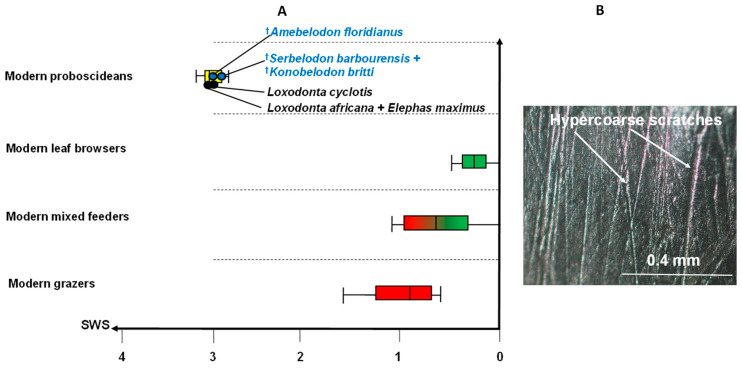
Scratch width scores for fossil and extant proboscideans and example of hypercoarse scratches found on proboscidean molar teeth. Box plot of scratch width scores for extant ungulates and proboscideans (**A**) and enamel surface of UF 38207 m2—*Serbelodon barbourensis* (lower second molar) showing the hypercoarse scratches typical of proboscideans and bark consumption (**B**). The box represents the central 50% of the scratch values and the bars represent the range (minimal and maximal values).

**Figure 7 biology-11-01748-f007:**
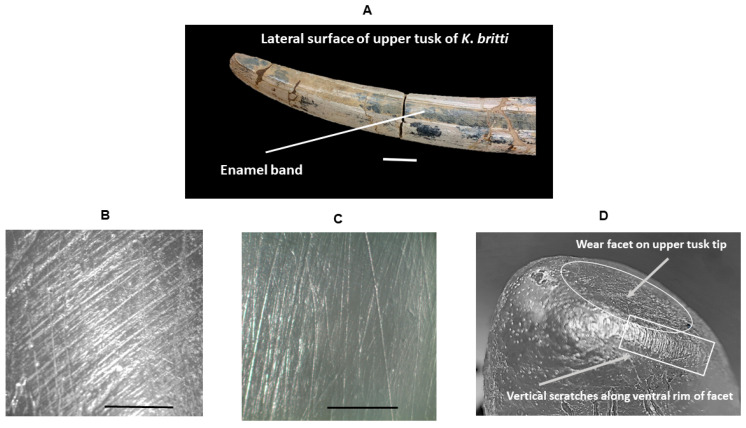
Enamel surfaces of upper shovel- tusker tusks. (**A**). Lateral enamel band found on shovel-tusker upper tusks (scale bar is 100 mm); (**B**). Ventral surface of right upper tusk of *K*. *britti* (UF 69998 at 50×). (**C**). Ventral surface of upper right tusk of *A*. *floridanus* (UF 212305 at 50×). 7D. Vertical scratches along the ventral rim of a wear facet on the upper right tusk tip of *A*. *floridanus* (UF 212305) The tusk tip in (**D**) is not magnified in the photo.

**Figure 8 biology-11-01748-f008:**
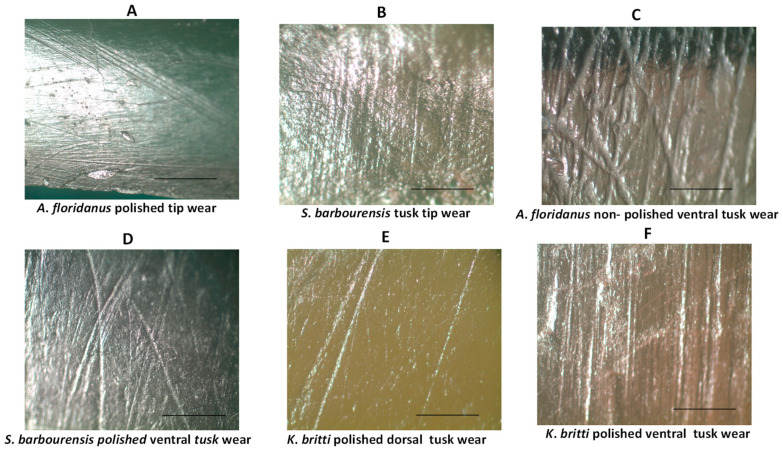
Examples of mandibular tusk wear scratch patterns. Enamel surface wear of shovel- tusker mandibular tusks. (**A**). Unusual mandibular tusk tip facet with randomly arranged scratches (*A*. *floridianus,* UF 217471); (**B**). *A*. *floridianus* (UF 212304) ventral mandibular tusk wear (non-polished); (**C**). *S*. *barbourensis* ventral mandibular tusk wear on wear facet (UF 39050); (**D**). *S*. *barbourensis* mandibular tusk tip wear (UF 97269; (**E**). *K*. *britti* dorsal mandibular tusk wear (UF 97269); (**F**). *K*. *britti* ventral mandibular tusk (UF 97269). Scale bar = 0.4 mm; magnification—(**A**). = 10×; (**B**–**K**) = 35×.

**Table 1 biology-11-01748-t001:** Molar specimens analyzed.

	Locality	Museum	Number
*A. floridanus*	Mixson’s Bone Bed; Alachua Formation; Todd Co.	AMNH	37
*A. floridanus*	Mixon’s Bone Bed; Alachua FM; Levy Co.	YPM	3
*A. floridanus*	Tyner Farm	UF	4
*A. floridanus*	McGehee Farm	UF	1
*S. barbourensis*	Love Bone Bed	UF	10
*K. britti*	Moss Acres Racetrack	UF	3

Species, fossil localities, collections sites, and number of specimens for fossil molars analyzed using microwear. AMNH = American Museum of Natural History in New York, USA; YPM = Yale Peabody Museum in Connecticut, USA; UF = University of Florida Museum of Natural History in Florida, USA.

**Table 2 biology-11-01748-t002:** Tusks Analyzed.

Tusks Analyzed			
Species	Locality	Number of Upper Tusks	Number of Lower Tusks
*A. floridanus*	Tyner Farm	5	3
*K*. *britti*	Moss Acres	4	1
*S*. *barbourensis*	Love Bone Bed	0	1

Species, fossil localities, and number of upper and lower tusks analyzed. All tusks are from the Florida Museum of Natural History.

**Table 3 biology-11-01748-t003:** Summary Microwear Statistics.

Taxon	Locality	N	*P*	LP	S	% F	%C	%M	%CH	%H	%G	%PP	SWS	0–17	Diet
*Amebelodon floridanus*	Mixson’s bone bed, Levy Co. (N = 43), Tyner Farm (N = 3), and McGehee Farm (N = 1)	47	36.4	2.3	10.1	0	4.3	2.1	93.6	0	6.4	8.5	2.9	97.9	B
*Serbelodonbarbourensis*	Love Bone Bed, Florida	11	27.6	6	12.6	0	0	0	100	0	18.2	9.1	2.9	100	B
*Konobelodonbritti*	Moss Acres Racetrack, Florida	3	21.2	1.3	10.3	0	0	0	100	0	33.3	0	2.7	100	B
*Loxodonta cyclotis*	Africa	6	29.8	17	12.9	0	0	16.7	66.7	16.7	33.3	33.3	2.8	83.3	B
*Loxodonta* *africana*	Africa	33	22.9	3.5	17.4	0	0	0	87.9	12.1	36.4	3.03	3.1	39.4	MF
*Elephas maximus*	Southeast Asia	10	20.9	3.7	18.3	0	0	0	90	10	50	0	3.1	40	MF

Summary statistics for molar microwear for the Florida shovel tuskers analyzed here. Key: N = number of specimens; *P* = average number of pits; LP = average number of large pits; S = average number of scratches; %F = percentages of individuals per taxon with predominantly fine scratches; %C = percentages of individuals per taxon with predominantly coarse scratches; %M = percentages of individuals per taxon with a mixture of fine and coarse scratches; % CH = percentages of individuals per taxon with a mixture of coarse and hypercoarse scratches; %H = percentages of individuals per taxon with predominantly hypercoarse scratches; %G = percentages of individuals per taxon with gouges; %PP = percentages of individuals per taxon with puncture- like pits; SWS = average scratch width score; and 0–17 = percentage of individuals per taxon with scratches that fall in the low scratch r.ange.

**Table 4 biology-11-01748-t004:** Upper tusk results for *A*. *floridanus*, *K*. *britti* and *S*. *barbourensis*.

Species	Locality	Number of Upper Tusks	Scratches Mostly Perpendicular to Enamel Band	Scratches Mostly Parallel to the Enamel Band	Tip Wear
*A. floridanus*	Tyner Farm	5	Yes (1 out of 5)	Yes (4 out of 5)	Yes (2 out of 5)
*K*. *britti*	Moss Acres	4	Yes (3 out of 4)	Yes (1 out of 4)	No

Upper tusk results for *A*. *floridanus*, *K*. *britti* and *S*. *barbourensis* showing the taxa, locality and number of tusks that could be examined for scratch patterns and the number of tusks out of the total number available for each taxon that had mostly perpendicular or parallel scratches in relation to the lateral enamel band as well as the presence of absence of tip wear.

**Table 5 biology-11-01748-t005:** Mandibular tusk wear results for *A*. *floridanus*, *K. britti* and *S*. *barbourensis*.

Species	Locality	Number of Lower Tusks	Dorsal and Ventral Wear Facets	Dorsal and Ventral Polishing	Tip Wear
*A. floridanus*	Tyner Farm	3	No	No	Yes (1 out of 3)
*K*. *britti*	Moss Acres	1	No	Yes	No
*S*. *barbourensis*	Love Bone Bed	1	Yes	No	Yes

Locality data, number or specimens, and examination results for *A*. *floridanus*, *K*. *britti* and *S*. *barbourensis* mandibular tusks.

## Data Availability

All data generated by this study are available in this manuscript and the accompanying Appendix A).

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
