# Peer review of "Dietary Habits and Tusk Usage of Shovel-Tusked Gomphotheres from Florida: Evidence from Stereoscopic Wear of Molars and Upper and Lower Tusks"

_biology, 2022, doi:10.3390/biology11121748_

Round 1
Reviewer 1 Report
Dear authors,
I would like to congratulate you for a carefully executed manuscript and the dental microwear of shovel-tusked gomphotheres. It is an important contribution in the field and significantly broadens our knowledge of the enigmatic group.
I have only very minor comments/suggestions for improvements, mainly regarding the size and readability of the figures (see annotated PDF).
The manuscript deserves to be published with very minor revisions and I expext this important contribution to be widely cited.

Author Response
We thank the reviewer. The suggestions were excellent and we have done everything that the reviewer suggested as seen below and in the revised attached manuscript
- Regarding the reviewers comment that Shovel tusker(s) is spelled with or without hyphen in the text. We have harmonized as shovel-tusker(s) throughout the document.
- For Figure 1, we have revised Figure 1 to make it much clearer and done as the reviewer requested and made the individual parts larger and even eliminated small text that wasn't necessary and even detracted from the clarity of the figure.
- We have revised Figure 2 to remove the provinces from the map and have used an outline map of Florida. The figure looks much better now.
- We have closed the parentheses after "small white pits per taxon").
- For Figure 3, we have replaced the missing line for the Y-axis and enlarged the print and uncorrupted the corrupted text.
- In Figure 5, we have eliminated "Platybelodon grangeri" in the text. It was an error.
- In Figure 6, we have replaced "Amebelodon cf. barbourensis" with Serbelodon barbourensis and replaced "Amebelodon britti" with Konobelodon britti. We have also indicated the dimensions of the scale bar and referenced the magnification in the revised caption.
- Figure 7 D is not magnified but the original size of the tusk. We have indicated this in the caption to avoid confusion.
- The labeling in Figure 8 (i.e., A, B, C, D, E, and F) has been inserted.
- We have corrected the font size for the word "plants" in line 477

Reviewer 2 Report
The manuscript is extremely interesting, as it assesses the paleodiet of a very interesting group of gomphotheres and that inferences from their diet were based on unscientific data. I believe that after a few adjustments, which I have highlighted in the attached pdf, the manuscript is ready to be accepted for publication. The results and conclusions are excellent and fill a long-awaited gap about the taxonomic group in question. I congratulate the authors and the journal for this important contribution to science. I would like to take this opportunity to put myself at your entire disposal for any doubts that may arise regarding my evaluation. Yours sincerely.
Author Response
We thank the reviewer for these excellent suggestions to improve the manuscript. We have made all of the corrections asked for by the reviewer in the revised manuscript (Please see attached):
In the Introduction:
- we have replaced "tetralophodon" with "tetralophodont" in line 37.
- we have inserted the reference asked for in line 41
- The paragraph in lines 61-73 has been moved to the Materials and Methods Section as asked and a simple sentence suggested by the author has replaced this paragraph in the Introduction.
- We agree with the reviewer that we should make it clear that there is a possibility that some or all of the gomphotheres studied here could have inhabited some locality at the same time and have inserted a sentence into the introduction in at the end of Section 1.2 to that effect.
- We have added the citations that the reviewer asked for in section 1.4
- In section 1.2 The reviewer asked how many of the dental specimens included in the sample were teeth from opposite sides of the same specimen. We checked an only found 2 specimens out of a total of that fit this description. We thank the reviewer for stating that this information may be optional for this manuscript because the effects on the results of only 2 specimens out of 58 would be negligible.
- In the Discussion, we have italicized the name "Konobelodon" twice as asked for by the reviewer.
- We have added the citation for "Browsers often exhibit a higher range of pit numbers from those taxa that are mixed feeders on grass and browse" as requested in the Discussion.

Reviewer 3 Report
A meticulous analysis of surface abrasions of molars and tusks of extinct gomphotheres. An impressive contribution to the arcane science of paleodiets of extinct species. Reconstruction of the dietary acquisition habits of gomphotheres reflect the changing environment of North America.
One small omission of the volume, page numbers of Reference 51 requires completion.
Author Response
We thank the reviewer for the encouraging words. We have included page numbers for reference number 51 in the revised manuscript.
